# Antiglycoxidative Properties of Extracts and Fractions from *Reynoutria* Rhizomes

**DOI:** 10.3390/nu13114066

**Published:** 2021-11-14

**Authors:** Arleta Dołowacka-Jóźwiak, Adam Matkowski, Izabela Nawrot-Hadzik

**Affiliations:** 1Department of Drug Form Technology, Wroclaw Medical University, 50556 Wroclaw, Poland; arleta.dolowacka-jozwiak@umw.edu.pl; 2Department of Pharmaceutical Biology and Biotechnology, Division of Pharmaceutical Biology and Botany, Botanical Garden of Medicinal Plants, Wroclaw Medical University, 50556 Wroclaw, Poland

**Keywords:** diabetes mellitus, *Reynoutria*, *Polygoni cuspidati rhizoma*, polyphenols, protein glycation, advanced glycation endproducts, fructosamine, protein oxidation, amyloid aggregation

## Abstract

Hyperglycemia, when sustained over a long time in diabetes mellitus (DM), leads to biochemical and cellular abnormalities, primarily through the formation of advanced glycation end-products (AGEs). In the treatment of diabetes, beside blood-sugar-lowering medications, a consumption of herbal products that can inhibit the AGEs’ formation is recommended. This study investigated the in vitro antiglycoxidative potential of extracts and fractions from the rhizomes of Japanese, Giant, and Bohemian knotweeds (*Reynoutria japonica* (Houtt.), *R. sachalinensis* (F. Schmidt) Nakai, and *R.*× *bohemica* Chrtek et Chrtkova). Their effects on glycooxidation of bovine and human serum albumin were evaluated by incubation of the proteins with a mixture of glucose and fructose (0.5 M) and 150 µg/mL of extract for 28 days at 37 °C, followed by measuring early and late glycation products, albumin oxidation (carbonyl and free thiol groups), and amyloid-β aggregation (thioflavin T and Congo red assays). The highest antiglycoxidative activity, comparable or stronger than the reference drug (aminoguanidine), was observed for ethyl acetate and diethyl ether fractions, enriched in polyphenols (stilbenes, phenylpropanoid disaccharide esters, and free and oligomeric flavan-3-ols). In conclusion, the antiglycoxidative compounds from these three species should be further studied for potential use in the prevention and complementary treatment of DM.

## 1. Introduction

Global prevalence of diabetes mellitus (DM) was 9.3% (463 million people) in 2019, and it is estimated to affect 10.2% (578 million) of the population by 2030 and 10.9% (700 million) by 2045. The incidence of DM is higher in urban (10.8%) than rural (7.2%) areas, and it is more common in high-income areas (10.4%) than in low-income countries (4.0%). Every second person with DM does not know about their disease [1]. Uncontrolled diabetes is associated with other diseases—cardiomyopathy, atherosclerosis, Alzheimer’s disease, neuropathy, nephropathy, and retinopathy. DM belongs to metabolic diseases. Diabetics have impaired sugar metabolism and insulin deficiency and/or decreased cellular sensitivity to insulin. Long-term hyperglycemia causes an increase in metabolic disorders and biochemical abnormalities; it results in increased protein glycation and accelerated development of complications: micro- and macroangiopathies [2,3,4]. The Maillard reaction is a biochemical transformation leading to non-enzymatic protein glycation and, consequently, to the loss or modification of their original role when occurring in the body [5,6,7]. The Maillard process is divided into three stages: early, intermediate, and late (Figure 1).

The early products of protein glycation include Schiff’s bases and the Amadori products. This reaction begins with the attachment of a reducing sugar carbonyl group (e.g., glucose, pentose, mannose, fructose, and galactose) to a free amino group of proteins, nucleic acids, or lipids to form an unstable compound called Schiff’s base (aldimine). This process is reversible [3,5]. This is followed by a slow rearrangement of the Schiff base, which gives rise to a stable ketoamine—the Amadori product. In vivo, Amadori products reach equilibrium after about 28 days, and by binding irreversibly to, inter alia, proteins, they begin to accumulate in cells, tissues, and organs. In an intermediate stage, the early products of the Maillard reaction are degraded. Through the oxidation and dehydration of Amadori products, various carbonyl compounds are formed (including glyoxal, methylglyoxal, 3-deoxyglucosone, N^ε^-(carboxymethyl)-lysine, and N^ε^-(carboxyethyl)-lysine). These products act as reaction promoters, reacting again with free amino groups, e.g., proteins. The last stage of glycation involves oxidation, dehydration, polymerization, cyclization, and condensation reactions with other amino groups. As a result, advanced glycation products, AGEs, are created [5,6]. These products are often fluorescent, insoluble, and irreversible compounds. In healthy people, modified proteins resulting from glycation are degraded and eliminated from the body in a physiological manner. It is worth mentioning that with the aging of the body or in the case of chronic diseases such as diabetes, atherosclerosis, and kidney disease, the glycation process is intensified, and the elimination of glycated proteins is significantly weakened. AGEs most often accumulate in the eyes, kidneys, atherosclerotic plaque, central nervous system, and brain, leading to disruption of the proper functioning of these organs and to micro- and macroangiopathy [3,5,6]. The glycation process occurs more intensively along with the oxidative process; both processes affect each other. It is the mechanism of the “vicious circle,” as a result of which the serious and negative consequences in the body worsen. Both of these accompanying processes are known as glycoxidation. Moreover, it has been shown that the glycation products that are formed during the Maillard reaction can be supplied with food. The content of glycated products depends on the degree of processing of the food and the way the meal is prepared. These products are also significantly associated with chronic diseases and their complications [3,5,6,7]

There are numerous studies searching for compounds that may reduce the concentration of AGEs and free radicals in the body by inhibiting their synthesis at different stages of the Maillard reaction [8,9,10]. The use of AGEs synthesis inhibitors is a promising therapeutic target. It was proved that both natural and synthetic compounds might act as inhibitors of AGEs, indicating different mechanisms of action [10]. Many synthetic compounds were tested for protein glycation inhibition. However, despite promising results, none of them were approved for the treatment of diabetic complications due to observed side effects [10]. Polyphenols, being naturally derived compounds found in plants, attract a large amount of attention from researchers because of their multidirectional biological activity, including significant antioxidant properties [8]. More and more studies on polyphenols demonstrate their beneficial anti-diabetic effects. Fernandez-Gomez et al. [11] conducted a study that sought new information concerning the inhibition of AGEs formation from extracts of coffee-roasting byproducts. It was concluded that the antiglycation activity of polyphenols contained in that extract might be related to their capacity to uptake carbonyl compounds and their capacity to react with the side chains of amino acid residues of proteins that block the glycation sites. *Myrica gale* extracts and rose extracts containing a high amount of polyphenols have a very high antioxidant capacity and significantly decreased α-amylase and α-glucosidase activities compared to acarbose [12,13]. Anthocyanidins and flavonols, which are the main components of berries and leaves, show a strong inhibitory effect on α-glucosidase, α-amylase, aldose reductase, and BSA glycation [12].

In the present study, we examined extracts from rhizomes of three species of large, invasive knotweeds from the genus *Reynoutria* Houtt. (Polygonaceae): *R. japonica* Houtt., *R. sachalinensis* (F.Schmidt) Nakai, and *R.* × *bohemica* Chrtek & Chrtková for their in vitro anti-glycoxidative activity. According to our previous research, these rhizomes are a rich source of polyphenols with high antioxidant capacity [14,15]. The *R. japonica* rhizomes (Huzhang in Chinese) are listed in the Chinese Pharmacopoeia, and they have been used for centuries in Chinese and Japanese traditional medicine for treating inflammation, jaundice, skin burns, scalds, and hyperlipidemia. Since 2017, under the synonym *Polygoni cuspidati rhizoma*, they have been included in the European Pharmacopoeia [15]. Despite not being a pharmacopoeial plant, another large knotweed—*R. sachalinensis* (Giant, or Sakhalin, knotweed)—has been, to some extent, used traditionally as an herbal medicine in East and North-East Asia for treatment of arthralgia, jaundice, amenorrhea, coughs, scalds and burns, traumatic injuries, carbuncles, and sores. In Europe, both species crossed and produced a hybrid called Bohemian knotweed (*R.* × *bohemica*). These plants are also occasionally used as wild spring vegetables both in their original range and in Europe [16].

Previous studies conducted on *R. japonica* leaf extracts indicate an antiglycation potential of these plants [17]. Moreover, studies in diabetic rats have shown that *R. japonica* rhizome extract reduced streptozotocin-induced early podocyte damage in the kidneys, and its active ingredient, emodin, inhibited methylglyoxal-mediated protein glycation [18]. Given the above, we have decided to investigate the antiglycoxidative potential of extracts and fractions from rhizomes of *R. japonica, R. sachalinensis*, and *R.* × *bohemica* [14,15]. BSA and HSA were used as model proteins for optimizing the method. Both BSA and HSA were the subject of glycation using a mixture of sugars (glucose and fructose), and the induced changes were analyzed in three steps: prevention of early and late formation of AGEs, albumin oxidation, and amyloid-β aggregation.

## 2. Materials and Methods

### 2.1. Reagents

The following chemicals were used in the study, all of which were of analytical grade if not mentioned otherwise: D (+)–glucose (C_6_H_12_O_6_, GLC), D (−)–fructose (C_6_H_12_O_6_, FRC): (both purchased from POCH Ltd. Gliwice, Poland); BSA: fraction V, >92% purity; M 66,430 Da, (Fluka, Buchs, Switzerland); HSA: fraction V, 98% purity, M 67,000 Da (this and all following reagents were from Sigma-Aldrich, St. Louis, MO, USA); *trans*-resveratrol (C_14_H_12_O_3_); aminoguanidine hydrochloride >98% purity; sodium azide (NaN_3_); dimethyl sulfoxide ((CH_3_)_2_SO, DMSO); 2,4-dinitrophenylhydrazine (DNPH); 5,5’-dithiobis (2-nitrobenzoic acid) (Ellman’s reagent, DTNB); Congo red; nitro blue tetrazolium chloride (NBT); thioflavin T; and trichloroacetic acid (TCA).

### 2.2. Preparation of Plant Extracts and Fractions

Plant extracts and fractions of *Reynoutria* rhizomes were prepared at the Department of Pharmaceutical Biology and Botany, Wroclaw Medical University, in accordance with the procedure described in the previous article [14]. The species were identified by Klemens Jakubowski from the Wroclaw Medical University Botanical Garden herbarium based on vegetative morphology and generative organs (according to available florae). All extracts and fractions were placed in glass bottles and stored until use at −80 °C. For testing, the extracts and standards (resveratrol and aminoguanidine) were dissolved in 50% DMSO at a stock concentration of 20 mg/mL [14,15].

### 2.3. In Vitro Glycation of Albumin

In our study, we used bovine serum albumin (BSA) and human serum albumin (HSA). BSA is the standard used in in vitro testing. The structure of BSA is well known, which allows it to be widely used in scientific research as a model protein for endogenous and exogenous ligand binding. The results of BSA glycation can be easily compared with other studies. However, in this study we also used HSA to confront the results with BSA glycation. BSA or HSA solutions were glycated according to the procedure described by Münch et al. in our own modification [19]. A mixture (0.5 mL, 0.5 M) of sugars (0.25 M glucose and 0.25 M fructose) was added to the BSA solution or HSA solution (0.5 mL, 10 mg/mL). A sample in which phosphate-buffered saline (PBS) buffer (0.5 mL) was added to the BSA solution or HSA solution (0.5 mL, 10 mg/mL) was the control of the reaction. The mixture of sugars (0.5 mL, 0.5 M) and 7.5 µL of the extract or fraction in question (150 µg/uL) were then added to the BSA solution or HSA solution (0.5 mL, 10 mg/mL). A sample in which 0.5 mL of phosphate-buffered saline (PBS) and 7.5 µL of the extract or fraction in question (150 µg/uL) were added to the BSA solution or HSA solution (0.5 mL, 10 mg/mL) was the control. The samples with glycation control contained sodium azide at 0.02%. All samples were performed in triplicate. The glycation process was performed at 37 °C for 28 days on the Heidolph Polymax 1040 Platform (Heidolph, Schwabach, Germany) using the Heidolph Incubator 1000 at approximately 50 rpm. After the incubation period was over, the dialysis of all solutions was performed against PBS buffer at pH 7.4 for 48 h at 4 °C. After the dialysis was over, the samples were divided at 1 mL of the analyzed material and frozen at −20 °C until fluorescence analysis and other determinations were performed.

### 2.4. Measurement of Fructosamine Levels

The method described by Johnson et al. was used for fructosamine measurement [20]. The glycated protein solution (20 µL) was added to 300 µM NBT solution (180 µL) prepared in sodium carbonate buffer (100 mM, pH 10.4). The solution was incubated for 30 min at room temperature. Absorbance was measured at 530 nm (µQuant UV/VIS spectrophotometer, Biotek, Winooski, VT, USA). Fructosamine levels were calculated using the standard 1-deoxy-1-morpholinofructose curve; the equations of the calibration curves and the R^2^ coefficients were y = 0.3459 × with R^2^ =0.9991 for BSA and y = 0.2516 × with R^2^ = 0.9995 for HSA; the values were expressed in mM/L.

### 2.5. Measurement of AGEs by Fluorescence

AGEs formation in glycated albumin samples was evaluated by the method previously reported by Münch et al. [19]. The fluorescence of glycated samples was measured at excitation and emission wavelengths of 370 nm and 440 nm, respectively, using the Perkin Elmer LS50B spectrofluorometer (Perkin-Elmer, Waltham, MA, USA). Results were expressed in arbitrary fluorescence units (AFU).

### 2.6. Measurement of Protein Carbonyl Groups

The method described by Rice-Evans et al. was used for the measurement of carbonyl groups in our own modification [21]. Glycated protein solution (100 µL) was added to 10 mM 2,4-dinitrophenylhydrazine (2,4-DNPH) (400 µL), mixed briefly, and incubated in the dark at room temperature for 60 min. Then, 30% (*w*/*v*) of trichloroacetic acid (TCA) (500 µL) and the solution were placed on ice for 10 min to enable proteins to precipitate. The precipitated protein was obtained after centrifugation at 10,000 rpm for 10 min at 4 °C. The residue was washed three times with 1 mL of 1:1 *v*/*v* mixture of ethanol (99.8%) and ethyl acetate, and then it was dissolved in guanidine hydrochloride (1 mL, 6 M). The absorbance of the solution was recorded at 365 nm. The concentration of carbonyl groups was calculated using the molar absorbance ratio for 2,4-dinitrophenylhydrazine (ε = 21 mM^−1^ cm^−1^) and expressed in mM/mg protein.

### 2.7. Thiol Group Estimation

To measure thiol groups, a method developed by Ellman was used with our own modification [22]. The following reagents were added to test samples: 100 µL of extracts using a stock concentration of 20 mg/mL and sodium-phosphate buffer at pH 8.0, with 800 µL each; 10% SDS (100 µL) and Ellman’s reagent (100 µL) each were used, while all reagents—except for the Ellman’s reagent—were added to the negative control sample. After all reagents were added, they were mixed and incubated for 60 min at 37 °C. The absorbance of the test samples was measured against a blank at λ = 410 nm. After the absorbance was obtained, a standard curve was prepared. The values of the equation of the curve and the R^2^ coefficient were y = 1.2418x with R^2^ = 0.9994 for BSA and y = 1.2492x with R^2^ = 0.9996 for HSA. The concentration of thiol groups was calculated using the molar ratio ε at 410 nm = 13.6 mM^−1^ cm^−1^; the results are expressed in mM/mg protein.

### 2.8. Determination of Amyloid-β Aggregation by Thioflavin T

The method described by Le Vine [23] was used for measurement of amyloid-β. Thioflavin T solution (32 μM) was mixed in NaOH buffer with 10% glycerol. A total of 300 μL of thioflavin T solution was added to 700 μL of the test sample; the whole mixture was mixed. The fluorescence was read after one hour of incubation. The excitation and emission wavelengths were set at λ = 440 nm (5 nm slit) and 450–650 nm (10 nm slit). The obtained reference spectrum was subtracted from the spectrum of the test sample. Results were expressed in Arbitrary Units (AU).

### 2.9. Determination of Amyloid-β Aggregation by Congo Red

Aggregation in the glycated sample was measured using Congo red in accordance with the method described by Klunk et al. [24]. A total of 100 μM of Congo red in PBS buffer at pH 7.4 was used with 10% ethanol. The glycated sample (500 µL) was incubated with Congo red solution (500 µL), and the absorbance was measured after incubation for 20 min at room temperature at 530 nm.

### 2.10. Statistical Analysis

Each assay was performed in at least five repetitions and presented as mean ± SD for *n ≥* 5. Statistical analysis was performed using GraphPad Prism v.9 software (GraphPad Software, San Diego, CA, USA). Initially, the Shapiro–Wilk test was used to assess the distribution of results. Significant differences between mean values were evaluated by two-way ANOVA and Tukey’s multiple comparisons test.

## 3. Results

This study evaluated the antiglycoxidative properties of rhizome acetone extract and four fractions (dichloromethane, diethyl ether, ethyl acetate, and butanol) of each Asian knotweed: *Reynoutria japonica*, *R. sachalinensis*, and *R.* × *bohemica*. The experiment was designed to evaluate the antiglycoxidative potential of each of selected extracts and fractions as well as two reference compounds: resveratrol and aminoguanidine. AGEs were determined—there was a measurement of the fluorescence value of total AGEs (AFU), fructosamine levels (mmol/L), as well as the level of free carbonyl groups (COOH) (mM/mg protein), free thiol groups (SH) (mM/mg protein), amyloid-β products–thioflavins T (AFU), and Congo red (nm). This allowed each test sample to be analyzed for antiglycoxidative efficiency in individual measurements. Time and temperature were constant parameters of the performed determinations. This study also aimed to check whether there was a difference in the glycation using various types of albumin (BSA and HSA). It was observed how the analyzed extracts or fractions affect glycation by inhibiting levels of AGEs, fructosamine, and amyloid-β products and how they affect oxidation of albumin by measuring thiol and carbonyl groups’ levels. The capability of the analyzed plant extracts/fractions to inhibit glycation was further considered by analyzing it at three levels: (A) (early and late) reaction of glycation, (B) albumin oxidation, and (C) amyloid-β aggregation.

### 3.1. Fructosamine Levels

After 28 days of incubation, fructosamine levels in glycated BSA were 176.63 mmol/L and, in HAS, 207.06 mmol/L. Those levels were significantly increased compared to native albumin (BSA 5.85 mmol/L and HSA 8.46 mmol/L). Appropriate modifications were applied for fructosamine levels to evaluate the contribution of the analyzed extracts and fractions to the glycation process. The results of determinations are shown in Figure 2 and Figure 3.

All acetone extracts significantly inhibited the fructosamine formation. However, the strongest inhibition was observed for *R. sachalinensis*. Ethyl acetate fractions had the most significant effect on fructosamine reduction, while diethyl ether fractions were slightly weaker. However, the best results were also obtained for *R. sachalinensis* (70% inhibition for ethyl acetate and 62% for diethyl ether). For butanol fractions, a clearly noticeable inhibition was only in *R. sachalinensis* (41% inhibition). Dichloromethane extracts, except for *R. sachalinensis* (39% inhibition), did not inhibit the fructosamine formation.

The same distribution of results was observed for HSA (Figure 3) as for BSA. In that case, *R. sachalinensis* extract and fractions of ethyl acetate (94% inhibition) and diethyl ether (41% inhibition) had the best results as well. *R. sachalinensis* ethyl acetate fraction lowered fructosamine levels more than reference compounds—resveratrol and aminoguanidine.

### 3.2. AGEs Levels

AGEs markers are heterogeneous molecules due to different structure and properties and can be divided into: (1) fluorescent and cross-linking compounds, e.g., pentosidine, 2-(2-furoyl) -4 (5)-(2-furanyl) -1H-imidazole, glyoxal-lysine dimer (GOLD), methylglyoxal-lysine dimer (MOLD), and (2) non-fluorescent and non-cross-linked compounds, for example, pyraline, Nε-carboxymethyl-lysine (CML), and N3- (carboxyethyl)-lysine. The formation of fluorescent AGEs during albumin glycation is generally evaluated by monitoring their fluorescence at excitation and emission maxima of 370 nm and 440 nm, respectively. There was a significant increase in the formation of fluorescent products within 28 days. After 4 weeks of incubation, the fluorescence intensity was significantly increased for glycated BSA (1321.64 AFU) and HSA (1551.11 AFU) compared to native BSA (43.83 AFU) and HSA (43.42 AFU), indicating the progressive formation of glycated AGEs. The presence of all acetone extracts significantly reduced the AGEs level. The results are shown in the Figure 4.

The results show the same trend as for fructosamine. *R. sachalinensis* had the strongest activity. It inhibited AGEs formation most potently among ethyl acetate fraction (86% inhibition), as well as among the most active acetone extract (84% inhibition) and diethyl ether fractions (80% inhibition). There was also significant inhibition for butanol fractions (60% inhibition) and dichloromethane fractions (35% inhibition). It should be added that despite the slightly weaker inhibition of the extracts and fractions of other species, the activity of some of them was exceptionally high, i.e., for *R.* × *bohemica,* these included the fractions of ethyl acetate (82% inhibition) and diethyl ether (60% inhibition) and the same fractions for *R. japonica*: ethyl acetate (76% inhibition) and diethyl ether (42% inhibition).

For HSA, there was a similar trend in the results as for BSA, but the inhibition values were lower than for BSA. This is illustrated in Figure 5.

### 3.3. Protein Carbonyl Group Levels

The addition of extracts or fractions during glycation significantly decreased the formation of carbonyl groups except for the dichloromethane fractions of *R. japonica*. The results are shown in Figure 6. Importantly, carbonyl group formation was most strongly inhibited by acetone extracts and fractions of ethyl acetate followed by diethyl ether fractions. However, *R. japonica* had the advantage in inhibition for ethyl acetate fractions (53% inhibition) and diethyl ether (41% inhibition) fractions.

For HSA protein, there was a similar trend among the extracts and fractions. *R. japonica* showed slightly stronger activity of extracts and for the most active fractions: ethyl acetate and diethyl ether. The results are shown in Figure 7.

The results showed that acetone extract of *R. japonica* and the most active fractions lowered the level of free carbonyl groups to the level of the tested control, similar to the reference compounds.

### 3.4. Thiol Group Levels

The evaluation of free thiol groups in BSA and HSA after glycation was performed using DTNB reagent. The number of free thiol groups in BSA and HSA significantly decreased—by 96% compared to the negative control for BSA (0.153 and 4.511 mM/mg protein) and for HSA in the positive control by 97% compared to the negative control (0.2 and 4.586 mM/mg protein).

All analyzed extracts and fractions showed significant effects on protecting thiol groups from oxidation. The best results were observed for the known antioxidant resveratrol. There was also a very strong protection of thiol groups for extract and fractions of *R. japonica*—42% protection for acetone, 39% for diethyl ether, and 34% for ethyl acetate. It should be mentioned that extract and some fractions of *R. japonica* contain much more stilbenes, including resveratrol, than *R.* × *bohemica*, while *R. sachalinensis* does not contain stilbenes at all. The results for BSA are shown in Figure 8.

Additionally, for HSA, there was the strongest protection of thiol groups against oxidation for acetone extract of *R. japonica* (32% protection), whereas among fractions it was ethyl acetate from *R. japonica* (45% protection). Resveratrol (40% protection) and aminoguanidine (36% protection) protected thiol groups at a similar level to extract and the ethyl acetate fraction of *R. japonica*, as shown in Figure 9.

### 3.5. The Effect of Extracts on Amyloid-β Aggregation Thioflavin T Assay

The fluorescence intensity of thioflavin T was significantly elevated in glycated albumin compared to the negative control, suggesting that glycation of protein gradually induced the formation of amyloid structure in BSA and HSA. The results obtained in this study for BSA showed that the presence of all extracts or fractions very strongly, by 60–88%, inhibited thioflavin T-labeled amyloid-β aggregation, with amyloid structure formation being most strongly inhibited by ethyl acetate fractions and least inhibited by dichloromethane fractions. The results are shown in Figure 10.

For HSA, the presence of all extracts also effectively inhibited albumin aggregation by 62–90%. In that case, ethyl acetate fractions showed the strongest inhibition as well, similar to resveratrol and aminoguanidine. A summary of the antiglycation activity inhibiting amyloid-β product formation with thioflavin T is shown in Figure 11.

### 3.6. The Effect of Extracts on Amyloid-β Aggregation Congo Red Assay

Analogous results were obtained when the structure of amyloid-β was analyzed using Congo red. Maximum absorbance was observed when BSA and HSA were incubated with the mixture of sugars. According to the results obtained in this study, the presence of extracts or fractions causes inhibition of the aggregation process. Ethyl acetate fractions and acetone extracts showed the strongest inhibition of amyloid-β formation (53–82%) when incubated with BSA and sugars, whereas the presence of dichloromethane and butanol fractions showed an inhibition of 4–49% (Figure 12).

For HSA, the presence of most extracts and their fractions effectively inhibited amyloid-β aggregation by 89–61%. The results for HSA are shown in Figure 13.

### 3.7. Total Antiglycation Potential of Plant Extracts

The total antiglycation potential of the analyzed extracts was indicated in accordance with their total efficiency in antiglycoxidative activity by inhibiting the formation of fructosamine, AGEs, amyloid-β products, carbonyl groups, and by protecting thiol groups. The corresponding ranking for each plant extract compared to BSA and HSA is shown in Table 1 and Table 2.

According to the above table, the analyzed extracts and fractions that inhibited glycation at the stage of early and late AGEs formation were the ethyl acetate fractions: *R. sachalinensis*, *R. japonica*, and *R.* × *bohemica*. The ethyl acetate fraction of *R. sachalinensis* showed even better activity than aminoguanidine. The diethyl ether fraction of *R. sachalinensis* and *R. japonica* also inhibited glycation more strongly than the respective acetone extracts. The ethyl acetate and diethyl ether fractions contain compounds with the highest antiglycation potential. The remaining butanol and dichloromethane fractions showed weaker activity than the primary acetone extracts.

Regarding the protection of thiol groups and inhibition of oxidation of carbonyl groups, the activity of the tested extracts and fractions decreased according to the order in which they are listed: *R. japonica* ethyl acetate, *R. japonica* acetone, *R. sachalinensis* ethyl acetate, and *R. japonica* diethyl ether. It is noticeable that fractions and extracts from *R. japonica* usually showed stronger antioxidant protection than other species. The high content of the antioxidants—stilbenes, including resveratrol in the extract and the most active fractions of *R. japonica*—may have contributed to this result.

In the third stage, aimed at hindering the aggregation of amyloid-β products, decreasing activity was observed according to the following order: resveratrol, aminoguanidine, *R.* × *bohemica* ethyl acetate, *R. sachalinensis* ethyl acetate, *R. sachalinensis* acetone, *R. japonica* ethyl acetate, *R. sachalinensis* diethyl ether, and *R.* × *bohemica* acetone. This shows a more significant effect of extracts and fractions extracted from *R. sachalinensis* and *R.* × *bohemica* than *R. japonica*, which may indicate a more significant effect of compounds other than stilbenes.

In summary, the highest inhibition of the glycation process at the stage of early and late product formation was recorded for: resveratrol, aminoguanidine, and *R.* × *bohemica* ethyl acetate together with *R. sachalinensis* ethyl acetate and, next, *R. sachalinensis* acetone, *R. sachalinensis* diethyl ether, and *R.* × *bohemica* acetone. As with BSA, ethyl acetate fractions were the most efficient inhibitors of in vitro glycation, with *R. sachalinensis* and *R.* × *bohemica* species doing so much more strongly than *R. japonica*.

Regarding the protection of thiol groups and significant reduction in the carbonyl group formation, the activity decreased in the following order: aminoguanidine, *R. japonica* ethyl acetate, *R. japonica* acetone, *R. sachalinensis* ethyl acetate, *R.* × *bohemica* ethyl acetate, and *R. japonica* diethyl ether. As with BSA, stronger activity was seen for extracts and fractions from the rhizomes of *R. japonica* species.

In contrast, at the level of the third step, the inhibition of amyloid-β aggregation activity decreased following the order: *R. sachalinensis* ethyl acetate, *R.* × *bohemica* ethyl acetate, resveratrol, aminoguanidine, *R.* x *bohemica* acetone, *R. japonica* ethyl acetate, *R.* × *bohemica* diethyl ether, and *R. sachalinensis* acetone. As with BSA, there was clearly stronger activity in extracts and fractions obtained from species other than *R. japonica*.

## 4. Discussion

The glycoxidation process causes serious changes in the structure of albumin. Albumin easily undergoes the process of glycation due to its long half-life (about 20 days); high content in the structure of cysteine, lysine, arginine, and free amino groups; as well as its strong nucleophilic properties [25,26]. In this study, the antiglyoxidative properties of extracts (acetone–water: 70% acetone) and fractions obtained by the fractionation of extracts: dichloromethane, diethyl ether, ethyl acetate, and butanol from three knotweed species: *Reynoutria japonica*, *R. sachalinensis*, *R.* × *bohemica* were investigated. All analyzed extracts and fractions were previously screened for antioxidant activity, and their phytochemical profile was also established [14,15]. The activity of the extracts and fractions was evaluated in three glycation steps: (1) glycation reaction (early and late), (2) oxidation of bovine or human albumin, and (3) aggregation of amyloid-β. In addition to the indicated extracts and fractions, as reference substances, we used a strong antioxidant (resveratrol) and a compound with known anti-glycation properties (aminoguanidine) [27]. One of the studied species, *R. japonica*, is a rich source of resveratrol, and its presence has been previously proven in the studied extracts and fractions [14,15]. A lower content of stilbenes, including resveratrol, was observed in extracts obtained from the rhizomes of *R.* × *bohemica.* In contrast, *R. sachalinensis* extracts and fractions did not contain resveratrol or any other stilbenes [14,15].

In the first step, glycation is the condensation reaction of the primary amino group of the amino acid residue of the protein with the free aldehyde or carbonyl group of the reducing sugars, resulting in the formation of N-substituted aldosylamine (also called Schiff’s base), which then forms Amadori products such as fructosamine. Fructosamine determination after albumin glycation is used to monitor the accumulation of early glycation products (Figure 14).

The lowest fructosamine levels were observed with *R. sachalinensis* extract (70% inhibition for BSA and 94% for HSA), followed by *R.* × *bohemica* (60% inhibition for BSA and 54% for HSA) and finally *R. japonica* (35% inhibition for BSA and 52% for HSA). A similar relationship was observed among the fractions. The strongest activity, stronger than acetone extracts, was observed for the ethyl acetate fraction of *R. sachalinensis*, which was slightly weaker for *R.* × *bohemica* and weakest for *R. japonica*. Slightly lower inhibition than acetone extracts was observed for the diethyl ether fractions. Dichloromethane fractions showed no inhibitory activity for HSA and slight inhibitory activity for BSA. Butanol fractions of *R. sachalinensis* showed significant activity (41% inhibition), but the other two species showed little activity. Similar results to those of the fructosamine assay were observed for the late stage of glycation, where total AGEs levels were tested. The correlation between the results of the early and late stages of glycation suggests that inhibition of the first stage may have a large impact on the latter. Presumably, the compounds contained mainly in the ethyl acetate fractions, and to a lesser extent in diethyl ether fractions, are responsible for the significant inhibitory activity of the acetone extracts. Previous studies show that these fractions, especially ethyl acetate, have the highest content of total polyphenols (583 mg/g (GAE) in *R. japonica*, 641 mg/g in *R. sachalinensis*, and 643 mg/g in *R.* × *bohemica*) and tannins (484 mg/g (GAE) in *R. japonica*, 528 mg/g in *R. sachalinensis*, and 511 mg/g for *R.* × *bohemica* ethyl acetate fractions) as well as the strongest antioxidant activity among all tested extracts and fractions [14]. Moreover, according to a previous study [14], ethyl acetate fractions contain the highest levels of procyanidins with a low degree of polymerization. Further, the ethyl acetate fraction of *R. sachalinensis* contained significantly more of these compounds than the same fraction of the other two species. The diethyl ether fractions of *R. sachalinensis*, while slightly less active, have the highest content of simple flavan-3-ols such as epicatechin, catechin, and epicatechin-3-*O*-gallate. Lower concentrations of these compounds were in diethyl ether fractions of *R. japonica* and *R.* × *bohemica*. Importantly, the diethyl ether and ethyl acetate fractions of the latter two species contain stilbenes (e.g., resveratrol), which are absent in *R. sachalinensis*, and it seems that these compounds do not play the crucial role in this aspect of antiglycation activity. Another important group of compounds observed at high concentrations in the most active fractions are phenylpropanoid disaccharide esters (hydroxycinnamic acids derivatives), such as vanicosides A and B. Their highest content was observed in the diethyl ether fraction of *R. sachalinensis* [14,15]. Hence, the sucrose hydroxycinnamic acids esters should also be considered as potential inhibitors of the formation of early and late glycation products. The results of other studies support our conclusions [28,29]. Xie et al. [30] summarized the structural features of flavonoids and procyanidins relevant to their antiglycation activity. They observed that glycosylation of hydroxylated flavonoids tended to reduce the inhibitory activity on AGEs formation, whereas dimers or trimers of procyanidins showed stronger inhibitory effects than catechins. Chen et al. [31], in turn, showed that catechin (CC) has a much greater effect in inhibiting AGEs formation than epicatechin (EC). CC was found to be more effective than EC in inhibiting RO^•^ and ^•^OH and showed a better inhibitory effect on β-glucosidase. Procyanidins extracted from cranberries inhibit methylglyoxal- and glucose-mediated glycation of human hemoglobin and human serum albumin [32]. The inhibitory effect of cranberry-derived procyanidins was partly related to their ability to intercept reactive carbonyls. Such effects have been attributed to procyanidin monomers, dimers, and trimers. Moreover, Sun et al. [33] demonstrated that procyanidins B1 and B2 were more effective inhibitors of protein glycation than their monomers (+)-catechins and (−)-epicatechin. The higher content of procyanidins with a low degree of polymerization in the tested ethyl acetate fractions than in the diethyl ether fractions (containing more, simple flavan-3-ols like catechin and epicatechin) may explain their stronger activity. So far, there are no studies showing the antiglycation activity of vanicosides A and B (belonging to the phenylpropanoid disaccharide esters) present in high concentration in the extract and diethyl ether fraction of *R. sachalinensis*. However, the high activity of these samples and the fact that these compounds contain anti-glycation-active hydroxycinnamic acids groups [28] encourage further research on these compounds.

Oxidative modifications of BSA and HSA during glycation were demonstrated by carbonyl and thiol group detection (Figure 15 and Figure 16) [34,35].

Oxidative processes play an important role in the formation of AGEs and can occur at least in two ways [34,35,36]. The first mechanism is the self-oxidation of free sugars in the presence of oxygen with the formation of reactive dicarbonyl forms that react with proteins to form ketoamines. The second mechanism is the oxidation of Amadori products. Reactive forms are then produced: protein enediols and protein dicarbonyls, generating the formation of AGEs [37]. Oxidation-induced carbonyl protein formation is also accompanied by the loss of free thiols in albumin [4]. In our study, the highest concentration of carbonyl groups was detected in glycated BSA and HSA (positive control). The addition of plant extracts during glycation significantly reduced the formation of carbonyl groups, indicating their antioxidant activity. The results of the antioxidant potency assessment showed that among the three tested extracts, the highest inhibition of carbonyl group formation in albumin was observed with *R. japonica* (48% inhibition for BSA and 39% for HSA), followed by *R.* × *bohemica* (34% inhibition for BSA and 37% for HSA), and finally *R. sachalinensis* (39% inhibition for BSA and 37% for HSA). Even stronger inhibition was observed with the ethyl acetate fractions of *R. japonica* (53% inhibition for BSA and 48% for HAS—comparable to the results for the diethyl ether fractions)*,* which was lower for *R. sachalinensis* and slightly weaker for *R.* × *bohemica*. Dichloromethane extracts showed the lowest inhibitory activity (2–20%). The results of the test assessing the oxidation of thiol groups were similarly distributed. The highest number of free thiol groups was observed with *R. japonica* extract (42% protection for BSA and 46% for HSA), followed by *R.* × *bohemica* (26% protection for BSA and 21% for HSA) and *R. sachalinensis* (17% protection for BSA and 25% for HSA). The strongest protection against oxidation was noted for the ethyl acetate fraction of *R. japonica*, less so for *R.* × *bohemica*, and only somewhat weaker for *R. sachalinensis*, whereas the diethyl ether fraction exhibited lower activity. The least protection was observed in the presence of dichloromethane fractions, which showed only a 4–8% increase in thiol groups. Moreover, among the samples tested, the potent antioxidant-resveratrol showed very strong antioxidant protection, manifested by the protection of thiol groups and the inhibition of carbonyl group formation. The high content of this compound in the extract and fractions of *R. japonica* could result in their stronger protection of albumin against oxidation compared to extracts and fractions from other species of knotweed with a lower concentration of this compound (*R.* × *bohemica)* or without it (*R. sachalinensis)*. Resveratrol has antioxidant and anti-glycation properties. Shen et al. [38] presented data on the dose-dependent antiglycation capacity of resveratrol, allowing it to bind methylglyoxal and form resveratrol–MG complexes. Arcanjo et al. [39] found that resveratrol significantly reduced the formation of carbonyl groups, protected thiol groups, and neutralized α-dicarbonyls. By acting as an antioxidant, resveratrol reduced the formation of AGEs. Yilmaz et al. [40] demonstrated that resveratrol, due to its effect in reducing the levels of reactive oxygen species (ROS), advanced oxidation protein products (AOPPs), and AGEs, among others, can be considered as a protective agent against MG-generated glyco-oxidative stress. Resveratrol can potentially reduce the toxicity of AGEs and inhibit glycation-induced complications [41]. The high protection against glycoxidation observed for extracts and fractions devoid of stilbenes (from *R. sachalinensis*) indicates the important influence of other compounds with proven antioxidant activity, such as the previously mentioned flavan-3-ols, procyanidins, and phenylpropanoid disaccharide esters [42,43]. Numerous studies confirm that the interaction of AGEs with RAGE receptors activates the formation of ROS and inflammation of the blood vessels. Additionally, also at this stage, the studied extracts can work as scavengers of free radicals [44,45,46].

The glycation process induces conformational changes in albumin by increasing the concentration of the amyloid-β structure, which plays a major role in albumin aggregation. The free thiol group at position Cys34 of albumin has strong nucleophilic properties and can be glycated. The Cys34, by reacting with reducing sugars, contributes to an increase in the level of β amyloid cross-structure in albumin. Proteins with α-helical structures undergo conformational changes into β-fold structures, which in turn aggregate into amyloid fibers [5,47]. Studies of HSA and BSA albumin showed that longer exposure of albumin to reducing sugars conforms the α-helix to a linear structure, forming amyloid-β [5,47,48]. Further, high plasma amyloid-β levels are associated with faster memory loss and an increased risk of developing Alzheimer’s and Parkinson’s diseases [48]. In our study, protein aggregation was assessed using the amyloid-β-specific pigments, thioflavin T and Congo red. The results for thioflavin T showed that for HSA and BSA, the level of amyloid structure was reduced by all extracts and fractions. Strong anti-amyloid properties were observed for all acetone extracts (ranging from 89 to 84% inhibition). Even stronger anti-amyloid properties were observed for the ethyl acetate fraction of all species. The diethyl ether fraction exhibited slightly weaker inhibition. The least protection of albumins was observed in the presence of dichloromethane and butanol fractions. A similar trend of results was observed with the Congo red assay. Here, too, the strongest inhibitory effect was observed for the ethyl acetate fractions of *Reynoutria* species. Two methods of detecting the amyloid-β structure are recommended to exclude false results, taking into account the limitations of the tests. Thus, the fluorescence of thioflavin T may be biased because polyphenolic compounds directly interact with thioflavin T, but Congo red is less sensitive in detecting the amyloid-β aggregate [26,49]. It is clear that the distribution of the results among the tested extracts and fractions for the inhibition of amyloid-β structure formation is very similar to the distribution of the results from the early and late stages of glycation. It means that the above-mentioned compounds had the greatest inhibitory effect on amyloid-β: flavan-3-ols, procyanidins with low degree of polymerization, and phenylpropanoid disaccharide esters and that inhibition of glycation by the test samples in the early and late stages resulted in less amyloid-β formation. Another mechanism for inhibiting amyloid-β formation is also possible, e.g., blocking albumin folding into the cross-β structure [26,50]; however, this should be determined in future research.

In all the tests carried out, dichloromethane and butanol fractions showed the weakest activity. Consistent with previous studies, the dichloromethane fractions contained the least amount of polyphenols and tannins and showed the weakest antioxidant activity. They are dominated by anthraquinones like emodin and physcion [14]. The mechanism of the anti-glycation action of these compounds is not well understood yet. It has been suggested that the anti-glycation activity of anthraquinones can be attributed to their ability to bind and stabilize the structure of albumin [51,52,53,54]. The relatively poor results for the butanol fraction may be surprising, taking into account the high content of procyanidins [14]. This fraction is dominated by procyanidins with a high degree of polymerization, which, according to some studies, have weaker antioxidant and anti-glycation properties than low-degree procyanidins [32,55,56,57,58,59,60,61]. Additionally, there may be nonspecific and permanent binding properties of procyanidins contained in the butanol fraction to albumin, which could have influenced the observed results. There is little information on the relationship between a high mean degree of polymerization (mDP) of procyanidins and antioxidant and antiglycemic activities [62,63]. However, the mechanism of inhibition of non-enzymatic glycation by procyanidins has not been thoroughly investigated and also requires further research.

The major limitation of the present study, which would have to be verified in the future, is that the in vitro conditions do not translate directly into the physiological situations in a living organism. In particular, the protein and sugar concentrations used in such assays are higher than in vivo, as are the active doses of test substances. Usually, plant polyphenols are absorbed and biotransformed differentially in the body; therefore, the composition of the ingested matrix can vary from what reaches the blood serum [64]. Hence, the present results warrant the next logical step, i.e., to test the bioavailability of the extract constituents and perform the cell-based and in vivo evaluation of the antidiabetic potential. Furthermore, the formulation studies were envisaged in order to develop a stable, safe, and bioaccessible natural preparation.

On the other hand, pleiotropic effects of dietary polyphenols are well known, and the unique composition and previously reported activities of *Reynoutria* plants may provide a good starting material for targeting multiple factors of diabetes pathogenesis [20,65,66,67]

## 5. Conclusions

In summary, the extracts and fractions from the studied invasive and medicinal knotweeds showed significant antiglycoxidative activity. The strongest inhibition of the formation of early and late glycation products and the formation of the amyloid-β structure were observed in the fractions with the highest content of:−simple flavan-3-ols, such as epicatechin, catechin, and epicatechin-3-*O*-gallate;−procyanidins with a low degree of polymerization;−phenylpropanoid disaccharide esters that dominated in the rhizomes of *R. sachalinensis*.

On the other hand, the strongest inhibition of oxidative modifications of BSA and HSA during glycation was observed for *R. japonica* fractions containing the highest amount of stilbenes, including the well-known antioxidant, resveratrol. Some of the tested fractions showed an activity similar to the reference, known antiglycation compound–aminoguanidine. By inhibiting the glycation process in the early and late stages, protecting proteins against oxidation and amyloid-β formation, the natural products from all three *Reynoutria* species could save cells from damage caused by long-term exposure to glucose and could have the potential to reduce the risk of diabetic complications.

## Figures and Tables

**Figure 1 nutrients-13-04066-f001:**
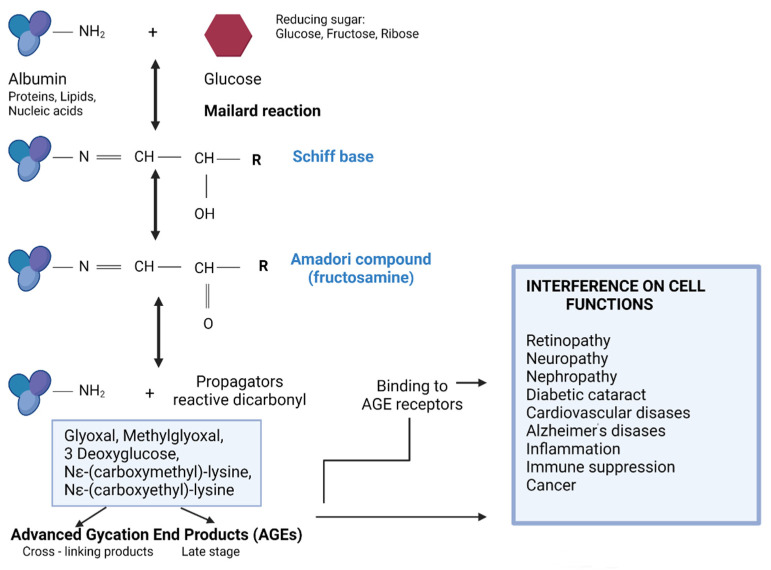
The three steps of a non-enzymatic glycation reaction and its consequences (the drawing was created in Biorender.com, accessed on 9 September 2021).

**Figure 2 nutrients-13-04066-f002:**
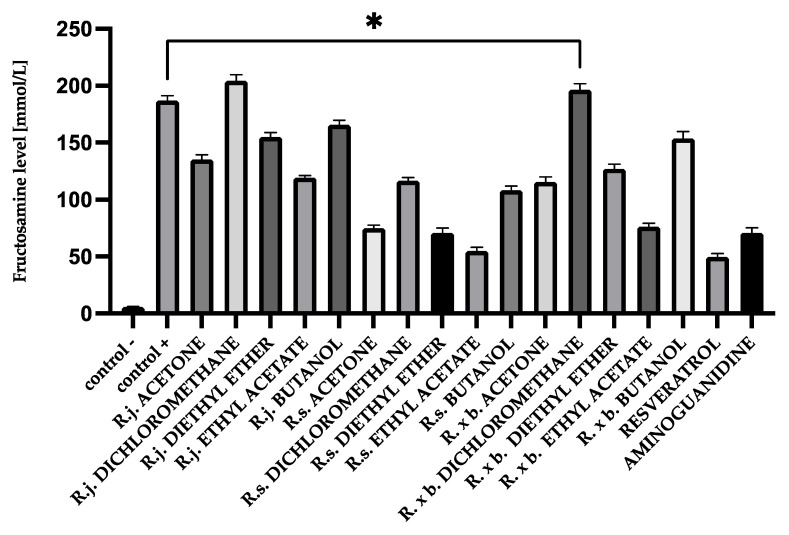
The effect of extracts and fractions on the in vitro formation of fructosamine in BSA glycation. R.j., R.s., and R. × b. mean *Reynoutria japonica*, *Reynoutria sachalinensis*, and *Reynoutria* × *bohemica*, respectively. Error bars shown in this figure are means ± SD for *n* ≥ 5. * Statistically significant at *p* ≤ 0.05 compared to control +. All other results are statistically significant at *p* ≤ 0.0001 compared to control +. Detailed information can be found in the Appendix A.

**Figure 3 nutrients-13-04066-f003:**
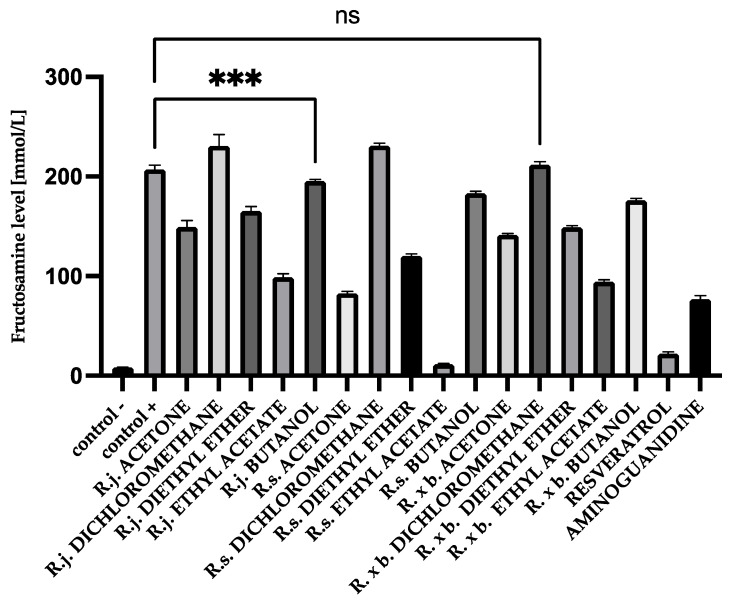
The effect of extracts and fractions on the in vitro formation of fructosamine in HSA glycation. R.j., R.s., and R. × b. mean *Reynoutria japonica*, *Reynoutria sachalinensis*, and *Reynoutria* × *bohemica*, respectively. Error bars shown in this figure are means ± SD for *n* ≥ 5. *** Statistically significant at *p* ≤ 0.001 compared to control +. ns—not statistically significant at *p* ≤ 0.05. All other results are statistically significant at *p* ≤ 0.0001 compared to control +. Detailed information can be found in the Appendix A.

**Figure 4 nutrients-13-04066-f004:**
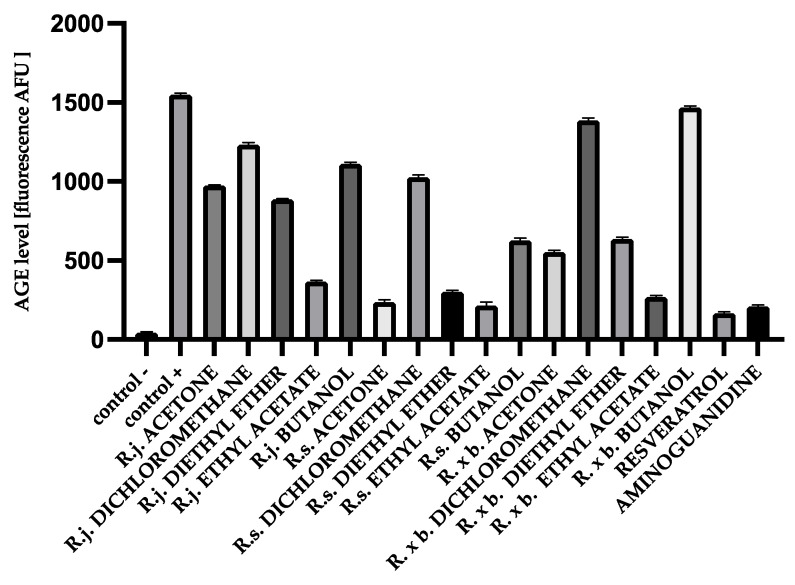
The effect of extracts and fractions on the formation of fluorescent AGEs in the in vitro BSA glycation. R.j., R.s., and R. × b. mean *Reynoutria japonica*, *Reynoutria sachalinensis*, and *Reynoutria* × *bohemica*, respectively. Error bars shown in this figure are means ± SD for *n ≥* 5. All results were statistically significant at *p* ≤ 0.0001 compared to control +. Detailed information can be found in the Appendix A.

**Figure 5 nutrients-13-04066-f005:**
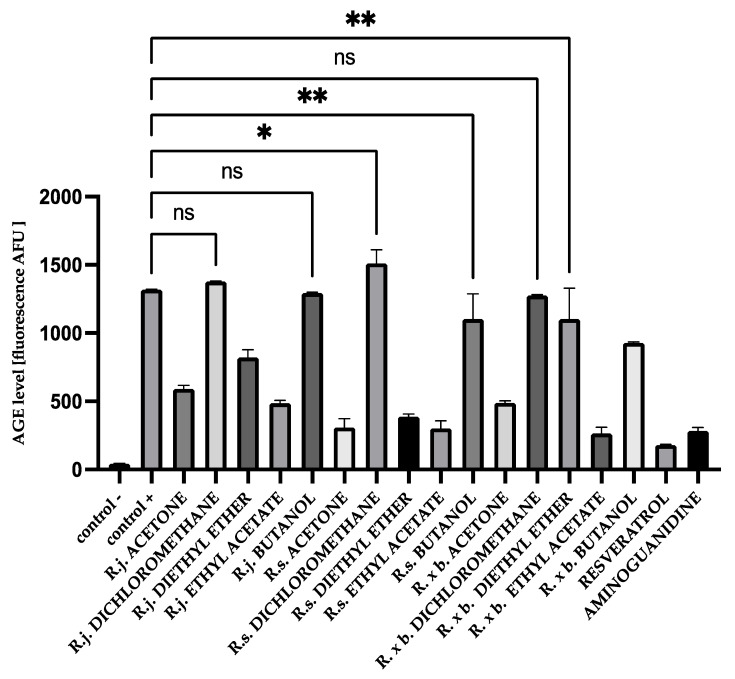
The effect of extracts and fractions on the formation of fluorescent AGEs in the in vitro HSA glycation. R.j., R.s., and R. × b. means *Reynoutria japonica*, *Reynoutria sachalinensis*, and *Reynoutria* × *bohemica*, respectively. Error bars shown in this figure are means ± SD for *n ≥* 5. ** Statistically significant at *p* ≤ 0.01 compared to control +; * statistically significant at *p* ≤ 0.05 compared to control +; ns—not statistically significant at *p* ≤ 0.05. All other results are statistically significant at *p* ≤ 0.0001 compared to control +. Detailed information can be found in the Appendix A.

**Figure 6 nutrients-13-04066-f006:**
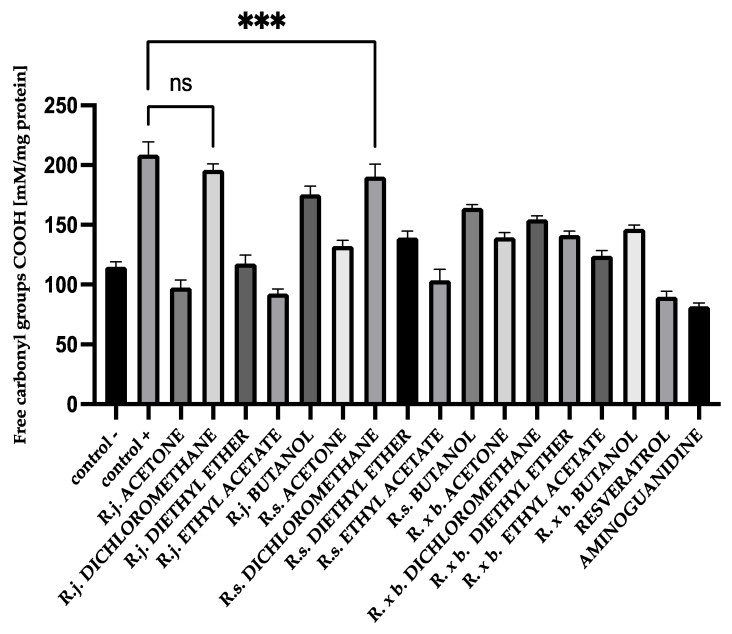
The effect of extracts and fractions on the in vitro formation of COOH groups in BSA glycation. R.j., R.s., and R. × b. mean *Reynoutria japonica*, *Reynoutria sachalinensis*, and *Reynoutria* × *bohemica*, respectively. Error bars shown in this figure are means ± SD for *n ≥* 5. *** Statistically significant at *p* ≤ 0.001 compared to control +; ns—not statistically significant at *p* ≤ 0.05. All other results are statistically significant at *p* ≤ 0.0001 compared to control +. Detailed information can be found in the Appendix A.

**Figure 7 nutrients-13-04066-f007:**
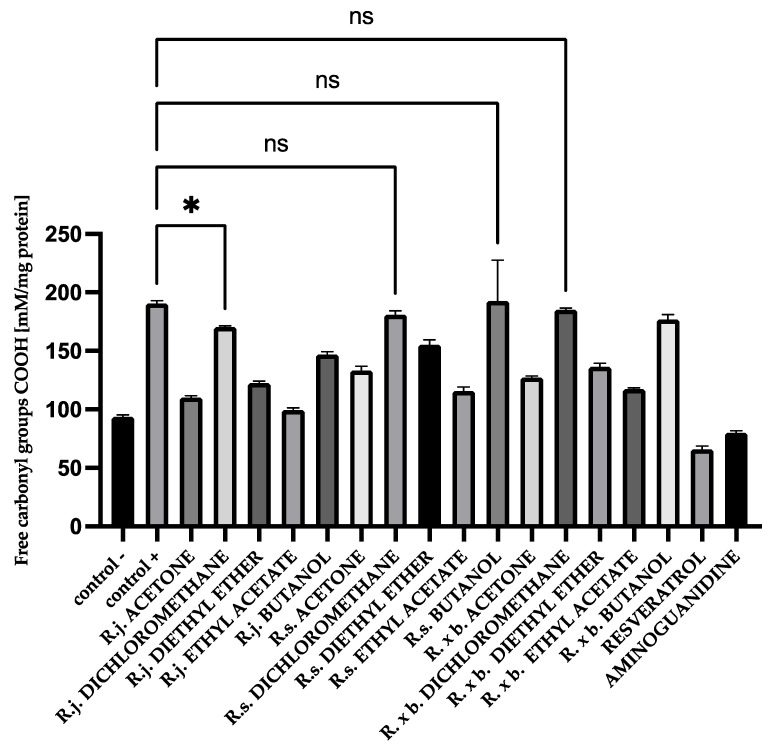
The effect of extracts on the in vitro formation of COOH groups in HSA glycation. R.j., R.s., and R. × b. mean *Reynoutria japonica*, *Reynoutria sachalinensis*, and *Reynoutria* × *bohemica*, respectively. Error bars shown in this figure are means ± SD for *n ≥* 5. * Statistically significant at *p* ≤ 0.05 compared to control +; ns—not statistically significant at *p* ≤ 0.05. All other results are statistically significant at *p* ≤ 0.0001 compared to control +. Detailed information can be found in the Appendix A.

**Figure 8 nutrients-13-04066-f008:**
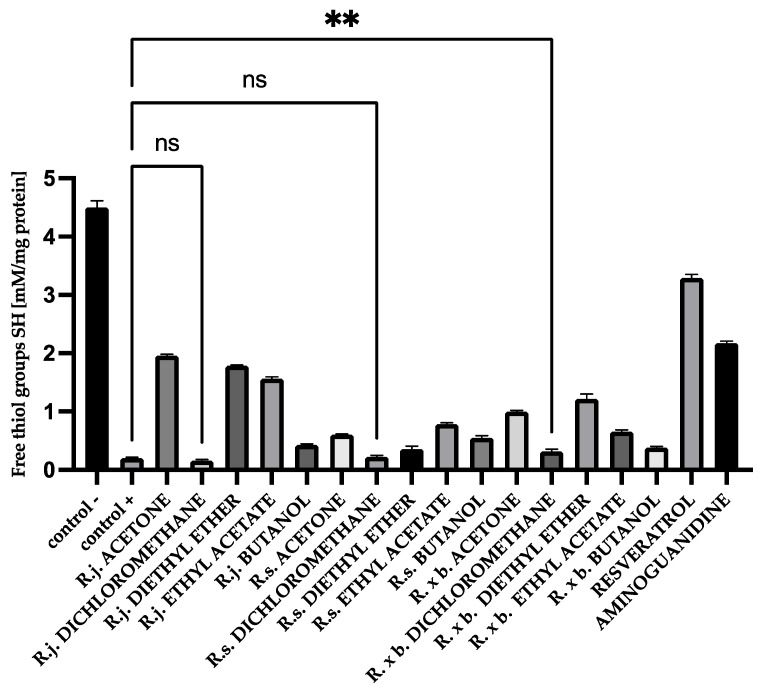
The effect of extracts and fractions on the in vitro formation of SH groups in BSA glycation. R.j., R.s., and R. × b. mean *Reynoutria japonica*, *Reynoutria sachalinensis*, and *Reynoutria* × *bohemica*, respectively. Error bars shown in this figure are means ± SD for *n ≥* 5. ** Statistically significant at *p* ≤ 0.01 compared to control; ns—not statistically significant at *p* ≤ 0.05. All other results are statistically significant at *p* ≤ 0.0001 compared to control +. Detailed information can be found in the Appendix A.

**Figure 9 nutrients-13-04066-f009:**
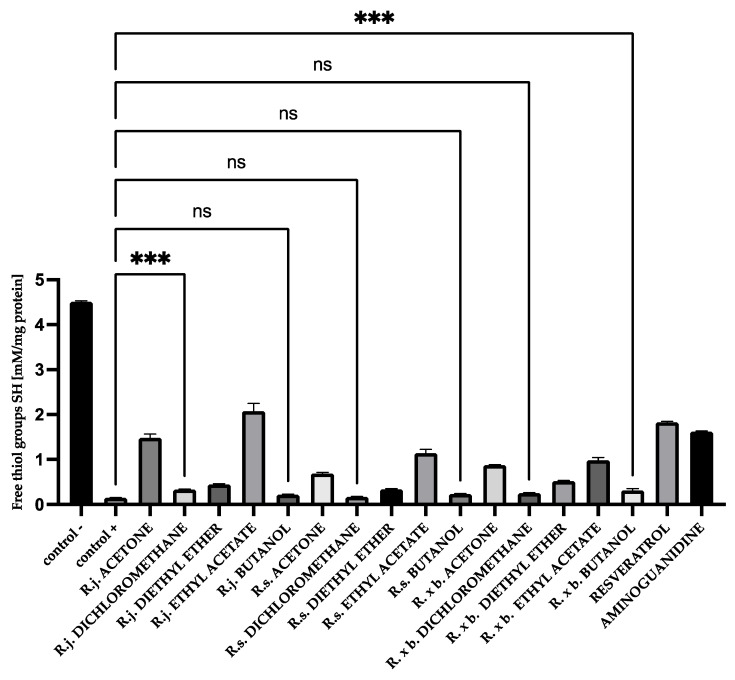
The effect of extracts and fractions on the in vitro formation of SH groups in HSA glycation. R.j., R.s., and R. × b. mean *Reynoutria japonica*, *Reynoutria sachalinensis*, and *Reynoutria* × *bohemica*, respectively. Error bars shown in this figure are means ± SD for *n ≥* 5. *** Statistically significant at *p* ≤ 0.001 compared to control; ns—not statistically significant at *p* ≤ 0.05. All other results are statistically significant at *p* ≤ 0.0001 compared to control +. Detailed information can be found in the Appendix A.

**Figure 10 nutrients-13-04066-f010:**
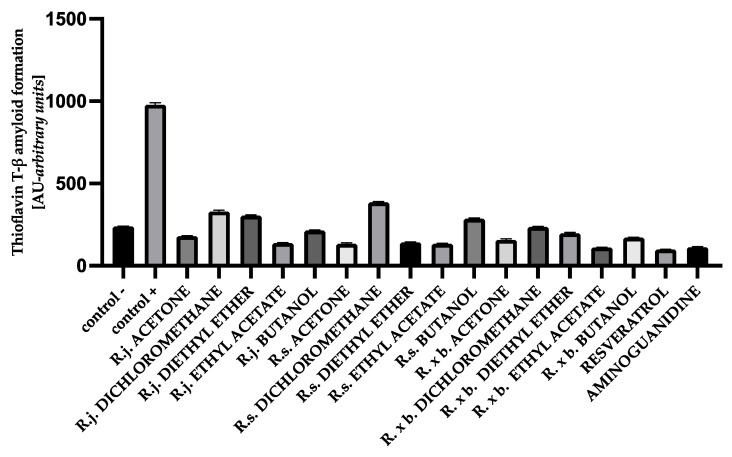
The effect of extracts and fractions on the in vitro formation of amyloid-β products—thioflavin T in BSA glycation. R.j., R.s., and R. × b. mean *Reynoutria japonica*, *Reynoutria sachalinensis*, and *Reynoutria* × *bohemica*, respectively. Error bars shown in this figure are means ± SD for *n ≥* 5. All results were statistically significant at *p* ≤ 0.0001 compared to control +. Detailed information can be found in the Appendix A.

**Figure 11 nutrients-13-04066-f011:**
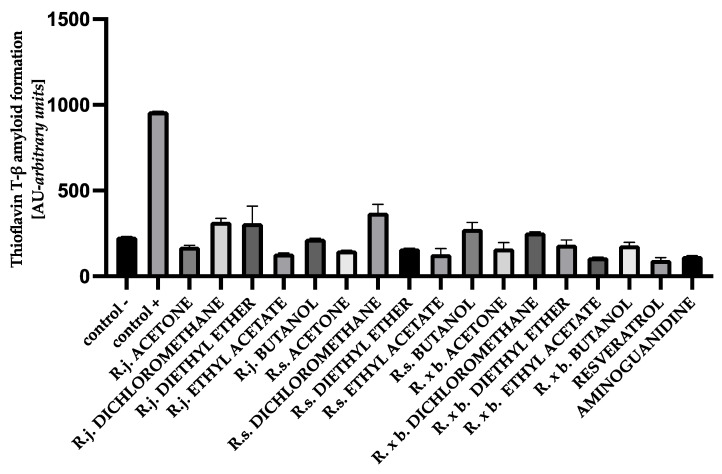
The effect of extracts and fractions on the in vitro formation of amyloid-β products—thioflavin T in HSA glycation. R.j., R.s., and R. × b. mean *Reynoutria japonica*, *Reynoutria sachalinensis*, and *Reynoutria* × *bohemica*, respectively. Error bars shown in this figure are means ± SD for *n ≥* 5. All results were statistically significant at *p* ≤ 0.0001 compared to control +. Detailed information can be found in the Appendix A.

**Figure 12 nutrients-13-04066-f012:**
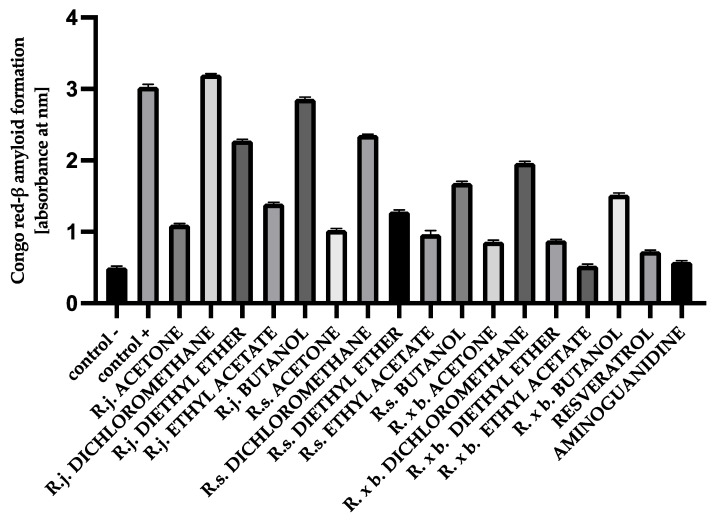
The effect of extracts and fractions on the in vitro formation of amyloid-β products—Congo red in BSA glycation. R.j., R.s., and R. × b. mean *Reynoutria japonica*, *Reynoutria sachalinensis*, and *Reynoutria* × *bohemica*, respectively. Error bars shown in this figure are means ± SD for *n ≥* 5. All results were statistically significant at *p* ≤ 0.0001 compared to control +. Detailed information can be found in the Appendix A.

**Figure 13 nutrients-13-04066-f013:**
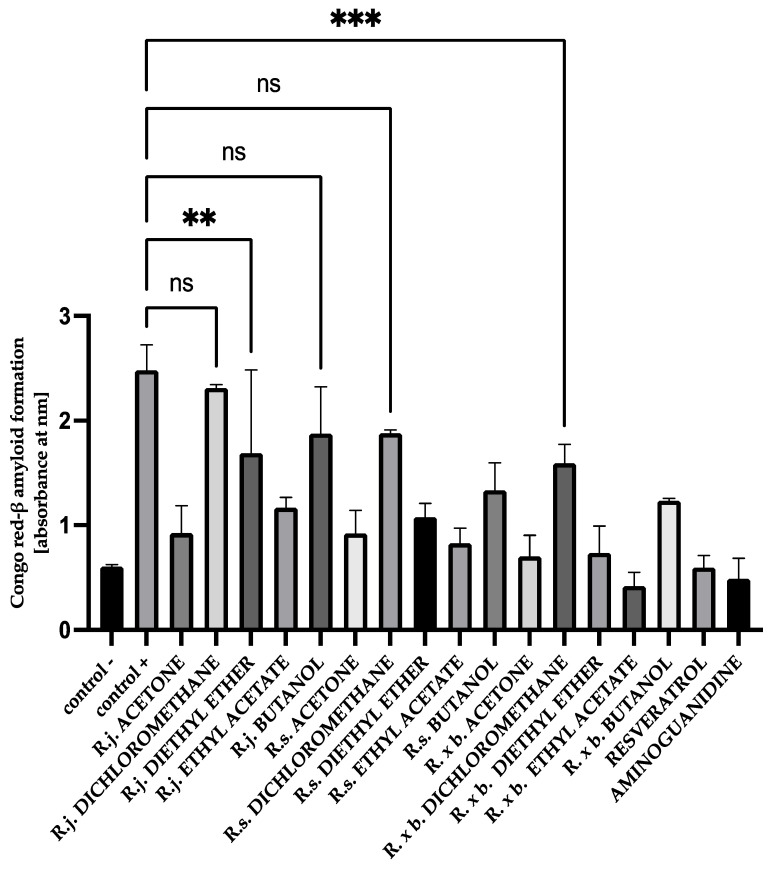
The effect of extracts and fractions on the in vitro formation of amyloid-β products—Congo red in HSA glycation. R.j., R.s., and R. × b. mean *Reynoutria japonica*, *Reynoutria sachalinensis*, and *Reynoutria* × *bohemica*, respectively. Error bars shown in this figure are means ± SD for *n ≥* 5. *** Statistically significant at *p* ≤ 0.001 compared to control, ** Statistically significant at *p* ≤ 0.01 compared to control; ns—not statistically significant at *p* ≤ 0.05. All other results are statistically significant at *p* ≤ 0.0001 compared to control +. Detailed information can be found in the Appendix A.

**Figure 14 nutrients-13-04066-f014:**
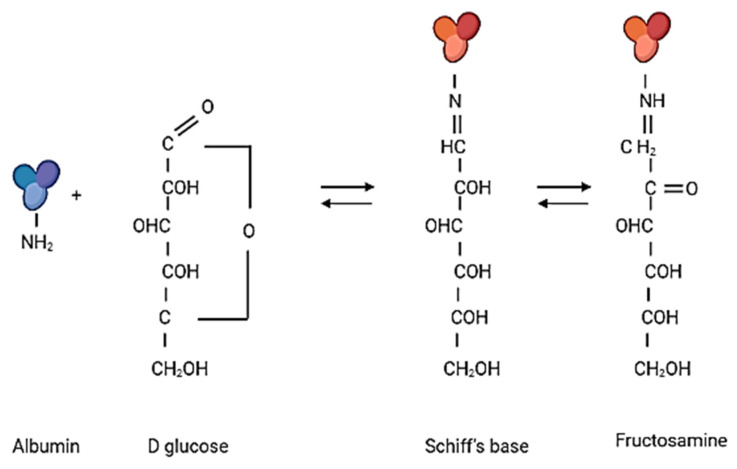
The first step of albumin glycation (the drawing was created in Biorender.com, accessed 9 September 2021).

**Figure 15 nutrients-13-04066-f015:**
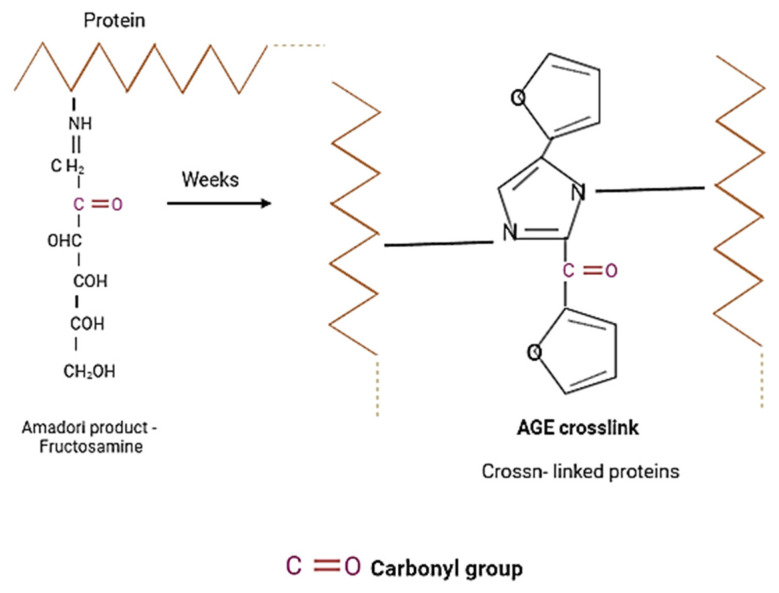
As a result of the glycation of albumin, the number of carbonyl groups increases. The Schiff’s base is changed into the more stable product Amadori (fructosamine), which is transformed into advanced glycation end-products (AGEs) through slow and complex reactions.

**Figure 16 nutrients-13-04066-f016:**
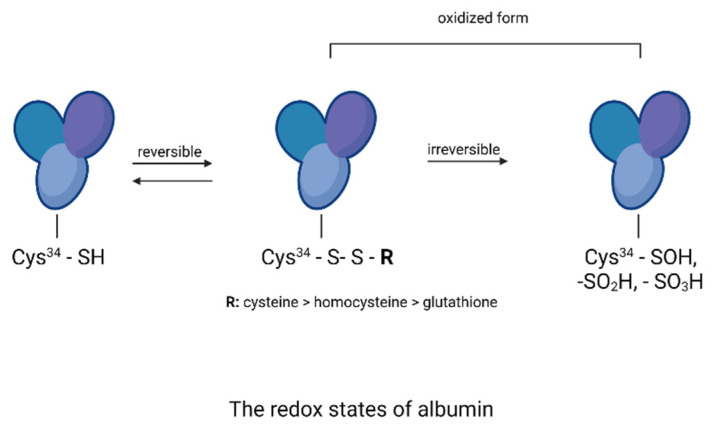
The redox states of albumin (the drawing was created in Biorender.com, accessed on 9 September 2021).

**Table 1 nutrients-13-04066-t001:** The summary of antiglycoxidation potential for each of the analyzed plant extracts and fractions at three stages of glycooxydation and their relative ranking (for BSA).

	Inhibition of Glycation		Inhibition of Protein Oxidation		Inhibition of Amyloid Aggregation	
Extracts and Fractions	Average Value (Fructosamines and AGEs)	Rank	Average Value (Protein Carbonyls and Thiols)	Rank	Average Value (Congo Red and Thioflavin T)	Rank
*R. japonica acetone*	391.7	12	34.3	4	62.6	10
*R. japonica dichloromethane*	720.3	17	68.4	16	115.6	16
*R. japonica diethyl ether*	366.1	11	42.4	6	107.1	15
*R. japonica ethyl acetate*	161.7	4	33.5	3	47.8	6
*R. japonica butanol*	452.9	14	61.1	15	75.73	12
*R. sachalinensis acetone*	112.4	7	46.4	8	46.6	5
*R. sachalinensis dichloromethane*	433.1	16	71.2	17	134.5	17
*R. sachalinensis diethyl ether*	147.1	6	48.5	9	49.9	7
*R. sachalinensis ethyl acetate*	80.8	2	36.5	5	46.1	4
*R. sachalinensis butanol*	281.2	10	56.9	14	99.8	14
*R. × bohemica acetone*	241.4	8	49.1	10	55.2	8
*R. × bohemica dichloromethane*	661.7	16	54.2	13	83.1	13
*R. × bohemica diethyl ether*	271.7	9	49.9	11	68.5	11
*R. × bohemica ethyl acetate*	129.1	5	43.6	7	42.8	3
*R. × bohemica butanol*	570.3	15	51.8	12	60.2	9
Resveratrol	65.6	1	29.7	1	34.1	1
Aminoguanidine	102.3	3	32.1	2	39.3	2

**Table 2 nutrients-13-04066-t002:** The summary of antiglycoxidation potential for each of the analyzed extracts and fractions at three stages of glycooxydation and their relative ranking (for HSA).

	Inhibition of Glycation		Inhibition of Protein Oxidation		Inhibition of Amyloid Aggregation	
Extracts and Fractions	Average Value (Fructosamines and AGEs)	Rank	Average Value (Protein Carbonyls and Thiols)	Rank	Average Value (Congo Red and Thioflavin T)	Rank
*R. japonica acetone*	259.2	10	34.7	4	57.3	10
*R. japonica dichloromethane*	553.1	17	53.1	13	93.0	17
*R. japonica diethyl ether*	277.8	11	38.2	7	63.2	11
*R. japonica ethyl acetate*	199.8	8	31.8	3	41.6	6
*R. japonica butanol*	502.9	16	45.4	11	68.1	13
*R. sachalinensis acetone*	90.1	5	41.4	9	46.3	8
*R. sachalinensis dichloromethane*	353.3	14	70.7	17	90.1	16
*R. sachalinensis diethyl ether*	116.9	6	48.6	12	50.9	9
*R. sachalinensis ethyl acetate*	80.8	4	36.1	5	23.7	1
*R. sachalinensis butanol*	324.4	12	61.1	15	74.0	14
*R. × bohemica acetone*	184.9	7	39.4	8	39.6	5
*R. × bohemica dichloromethane*	456.8	15	61.1	16	79.6	15
*R. × bohemica diethyl ether*	206.5	9	42.5	10	42.4	7
*R. × bohemica ethyl acetate*	80.4	3	36.5	6	34.2	2
*R. × bohemica butanol*	336.6	13	55.7	14	63.9	12
Resveratrol	60.5	1	20.6	1	35.5	3
Aminoguanidine	76.1	2	25.4	2	36.3	4

## Data Availability

The data presented in this study are available in the Appendix A here.

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
