# Peer review of "Antiglycoxidative Properties of Extracts and Fractions from Reynoutria Rhizomes"

_nutrients, 2021, doi:10.3390/nu13114066_

Round 1

Reviewer 1 Report

This study explores antiglycoxidative potential of Reynoutria japonica, R. sachalinensis, and R. x bohemica rhizome extracts and fractions. The work is quite interesting, clear, and well-organized. The introduction provides enough background. The aim of the study is clear. The methods are described in detail. The Results and Discussion are correctly described. I only have some minor observations that are listed in the following lines.

  1. The work would benefit from close editing.
  2. Figure 1: some parts are unreadable.
  3. Lines 288 and 290: Avoid using references in the results section.
  4. Add a list of abbreviations.

Author Response

Dear Reviewer,

We sincerely appreciate the positive and constructive comments and we have corrected the manuscript accordingly. The corrections are visible in the 'track changes' function of the revised Word document.

Specifically, we  (1) have performed a thorough proofreading and rewrote some phrases in the text when necessary; (2) we have replaced the Figure 1 image with a high-res graphics;  (3) the citations have been removed from the Results section. (4) Finally, the list of frequently used abbreviation was provided on the request of the Reviewer and the remaining (mostly of reagents) abbreviations were explained in the text.

Moreover, some other minor corrections were done:

  • an update of the authors' affiliations
  • a correction of wrongly numbered references [55, 56, and 57] and addition of one reference [57] that was missing from the original submission;
  • the spelling mistakes in Figure 16 have been corrected (albmin, glutatione)

We do hope that the manuscript has been sufficiently improved

sincerely,

the authors

Reviewer 2 Report

The manuscript presents all the necessary characteristics to be accepted for publication. It is easy to read and all sections are correctly written. It is easily observed that the authors are experts in this line of research, where they know well the species studied and the methodology used.

I emphasize in this work that the species have been correctly identified in the methodology. On many occasions, phytochemical work does not carry out this section correctly, and in those cases the results are more than debatable.

Author Response

Dear Reviewer,

We would like to thank for the positive and constructive comments. Indeed, we did our best to provide an insightfully interpreted and comprehensive study. We are now submitting the revised version which has been carefully proofread, resulting in some phrasing modifications to the main text. The corrections are visible in the 'track changes' function of the revised Word document.

Specifically, we have replaced the Figure 1 image with a high-res graphics, performed a thorough proofreading and rewrote some phrases in the text when necessary. Finally, the list of frequently used abbreviation was provided on the request of the other Reviewer and the remaining (mostly of reagents) abbreviations were explained in the text.

Moreover, some other minor corrections were done:

  • an update of the authors' affiliations;
  • the citations have been removed from the Results section;
  • spelling mistakes in Fig 16 have been corrected;
  • a correction of wrongly numbered references [55, 56, and 57] and addition of one reference [57] that was missing from the original submission;

We do hope that the manuscript has been sufficiently improved

sincerely,

the authors